Obsessive Compulsive Disorder; Bibliometric analysis; Mental health

**Corresponding author:**
Latha Ganti;
Email: latha_ganti@brown.edu

# Bibliometric analysis of OCD prevalence in youth populations of developing countries

Jacob Blaney[1], Sanjana Konda[2] and Latha Ganti[2,3] 

[1]Winter Park HS, Winter Park, FL; [2]The Warren Alpert Medical School of Brown University, Providence, RI, USA and [3]Orlando College of Osteopathic Medicine, Winter Garden, FL, USA

## Abstract

This paper is a bibliometric analysis of research of adolescent obsessive-compulsive disorder (OCD) in developing nations. An analysis of 4,807 papers was conducted to show trends in these areas. The most significant research came from developed countries – with the United States and England having the most publications and the strongest citation strength. However, developing countries play an important role in the development of OCD research because of how they deliver different perspectives into the field given their more distant associations with developed nation's research. This study will use multiple indicators of bibliometrics, most notably bibliographic coupling and citation strength, to draw conclusions to show the various contributions of different nations to the field of adolescent OCD.

## Impact statement

Bibliometrics are the quantitative analysis of academic publications. Using academic publications as a data source, bibliometric analysis attempts to provide a better understanding of how research is produced, organized and interrelated This study sheds light on the state of publications on the topic of obsessive-compulsive disorder and reports that developing countries are starting to publish more in this field, which is invaluable for the disease to be better recognized and managed.

## Introduction

Obsessive-compulsive disorder (OCD) is a mental disorder characterized by intense obsessions, followed by compulsions. Obsessions are considered unhealthy preoccupations that are difficult to control and cause great stress to individuals with OCD. To alleviate this stress of obsessions, they perform compulsions, which feel necessary (Brock and Hany, 2022). Individuals with OCD often exhibit greater activity in the frontal regions of the brain, likely causing their intrusive thoughts and stress (Pittenger, 2014). The neurobiological cause of OCD is often attributed to chemical imbalances of neurotransmitters, such as serotonin and glutamate (National Institute of Mental Health, 2023).

Approximately 1.2% of US adults have been diagnosed with OCD (Ruscio et al., 2010). It frequently occurs alongside other mental disorders, such as generalized anxiety disorder and specific phobias. OCD is diagnosed often in youth phases of life (The American Academy of Child and Adolescent Psychiatry, 2018). Males often show symptoms before the age of 10, while females will exhibit them later in adolescence (Stein et al., 2019). Family history, past history of trauma, anxiety, depression and neurobiological differences in the brain are all risk factors for OCD (Brander et al., 2016). In addition, the peripartum and postpartum stages for women exhibit an increased risk for OCD (Fairbrother et al., 2021).

Patients with OCD often present with frequent checking behaviors, significant and obvious anxiety, and repetitive actions that reflect compulsions. The patient may be aware of the irrationality of their thoughts, but they feel compelled to give in to their compulsions (Brock and Hany, 2022). Treatment of OCD involves a combination of cognitive therapy or selective serotonin reuptake inhibitor medications, and in extreme cases, electroconvulsive therapy (Liu et al., 2014; Maraone et al., 2021).

Bibliometric analysis is helpful for understanding the environment that exists around OCD. It highlights publication trends, countries of origin and citation networks to evince patterns in OCD research. This paper aims to visualize and analyze these trends to provide insights into OCD research.

## Materials and Methods

### Data

The data used in this bibliometric analysis are from the Web of Science Core Collection. TS = ((obsessive-compulsive disorder OR obsessive compulsive disorder OR OCD) AND

(teenagers OR 12–17 OR youth OR adolescent OR high school)) was the search strategy that was employed. A total of 4,807 articles were finally retrieved for analysis.

## Methods

The main indicators used for evaluations include the number of published articles from countries, institutions and authors over the years, specifically relating to OCD prevalence in youth. VosViewer is the principal tool of analysis, with analysis strength being the main method for association. VosViewer was particularly useful in this bibliometric analysis as it provided a robust tool to visualize and identify association strengths and trends across large datasets effectively (Arruda et al., 2022).

## Results

Figure 1 is a visualization of the citation strength of publications from specific countries. Figure 2 is a visualization of the publication strength from specific countries. In both figures, the United States, England and Canada have the strongest citation and publication strength. This reflects the importance of these countries in adolescent OCD research. Articles, on average, are skewed toward 2012.

Figures 3 and 4 break down the bibliometric visualization into developed and developing countries, respectively. Figure 3 reflects an interesting pattern – countries that have traditionally been first world for a very long time hold close links in OCD research, such as the United States and England, whereas nations like Sweden and Norway have more distant links. Turkey (including Turkiye), India and the People's Republic of China (PRC) have the strongest citation strengths among developing countries.

## Organizations

Through compiling various organizations, it can be shown that the greatest number of documents published and cited are from the University of Florida [Figure 5]. The next 10 organizations originate in the United States as well. The top few organizations have strong links between them, reflecting a strong collaboration between them.

## Discussion

This study compiled 4,807 publications published in the period 2012–2020. The countries with the strongest citation strengths were the United States, England and Canada. Among developing countries, the PRC, India and Turkey had the strongest citation strengths. As shown by Figure 6, the publications by most developed countries, such as the United States and Canada, are older. Meanwhile, most newer documents come from developing or recently developed nations. This is very significant especially given how some of these nations have cultures that traditionally stigmatized OCD. Qatar, as a primary example, has a significant issue with negative views towards those with OCD (Kehyayan et al., 2021). However, it is seen from Figure 5 that Qatar has produced some of the newest publications on the research of youth OCD. This trend reflects how these countries are modernizing and becoming more aware about the problems of OCD. Developing countries have shown that they are invaluable in OCD research because of the different perspectives that they offer. The above charts generally reflect distant associations between developed and developing country research. This illustrates a more multifaceted approach to research in OCD and helps ensure OCD research is in depth. Within developed countries, nations such as the United States and

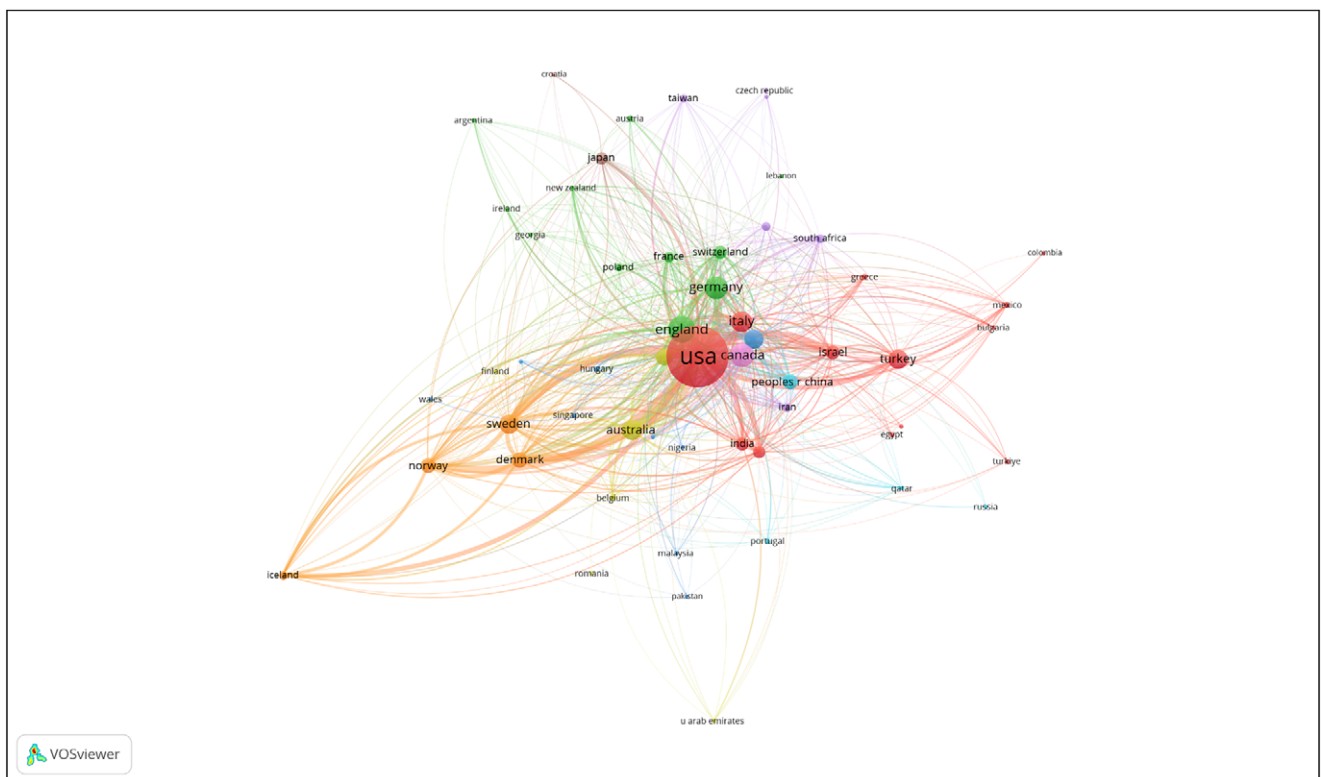

**Figure 1.** Visualization of the citation strength of publications from specific countries.

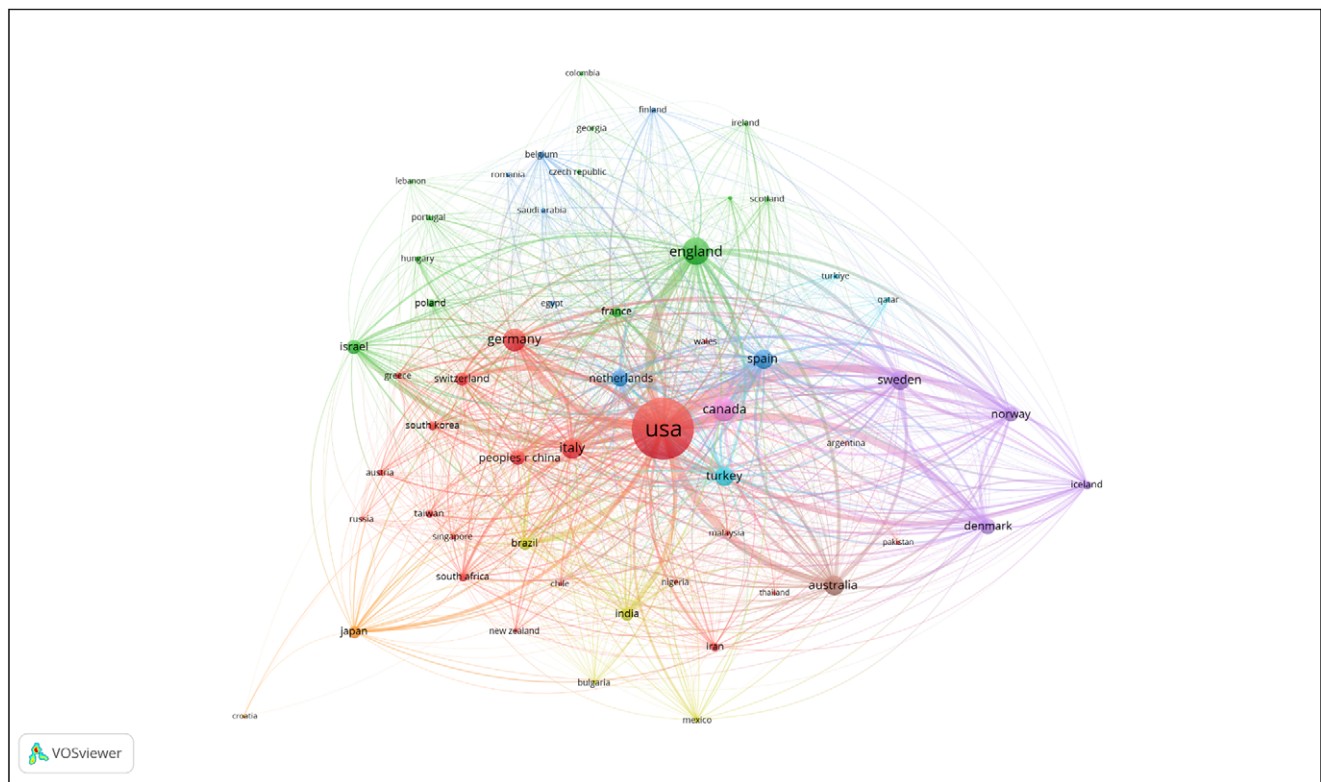

**Figure 2.** Bibliograph of countries. Size of circle indicates number of publications, lines represent association strength.

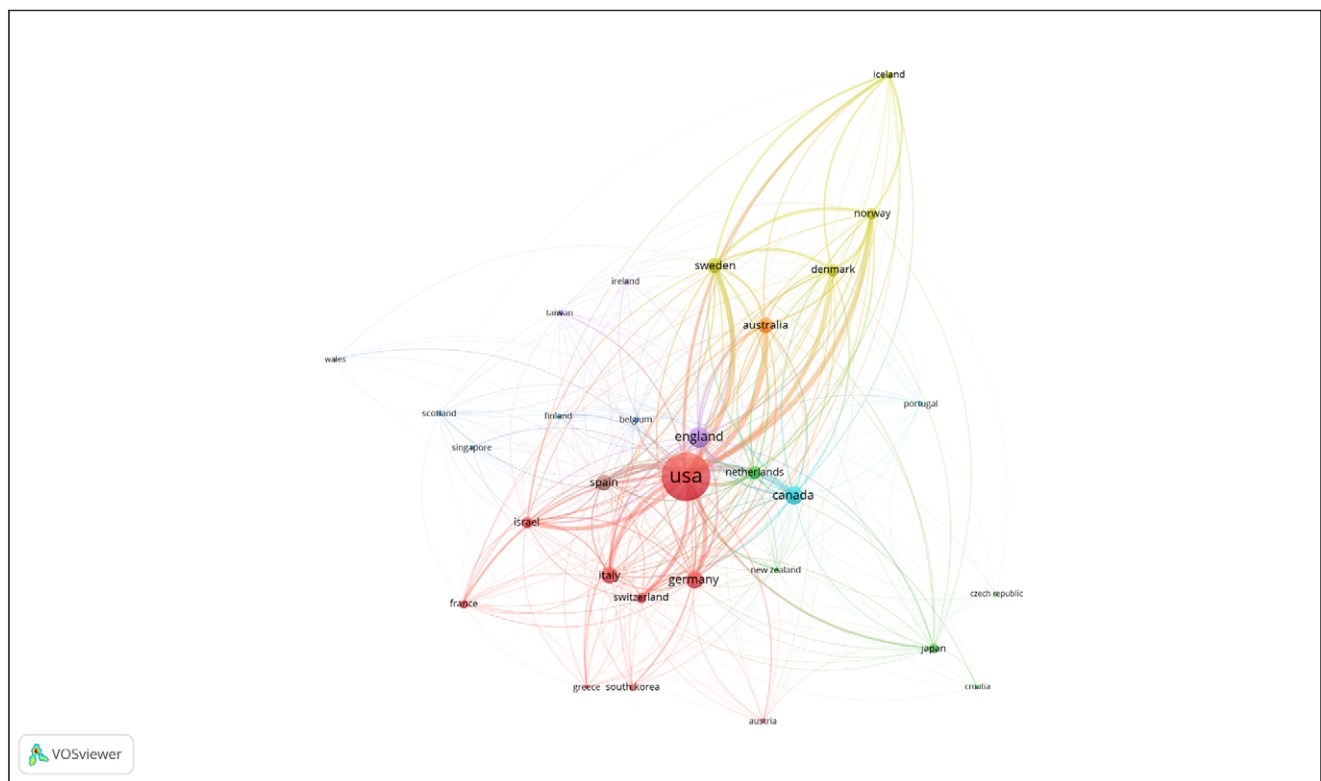

**Figure 3.** Countries considered to be developed by the International Monetary Fund (IMF).

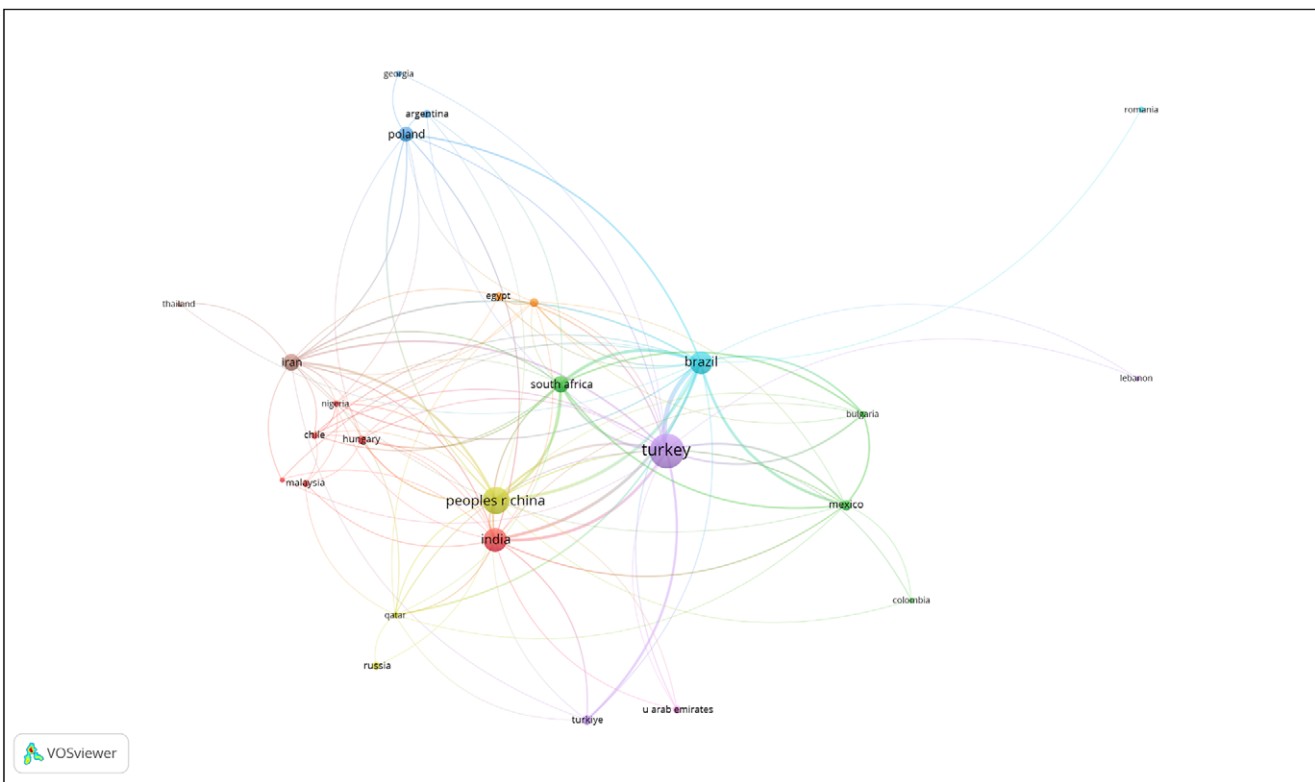

**Figure 4.** Countries considered developing by the IMF. *Note:* Turkey and Turkiye are considered to be one entity.

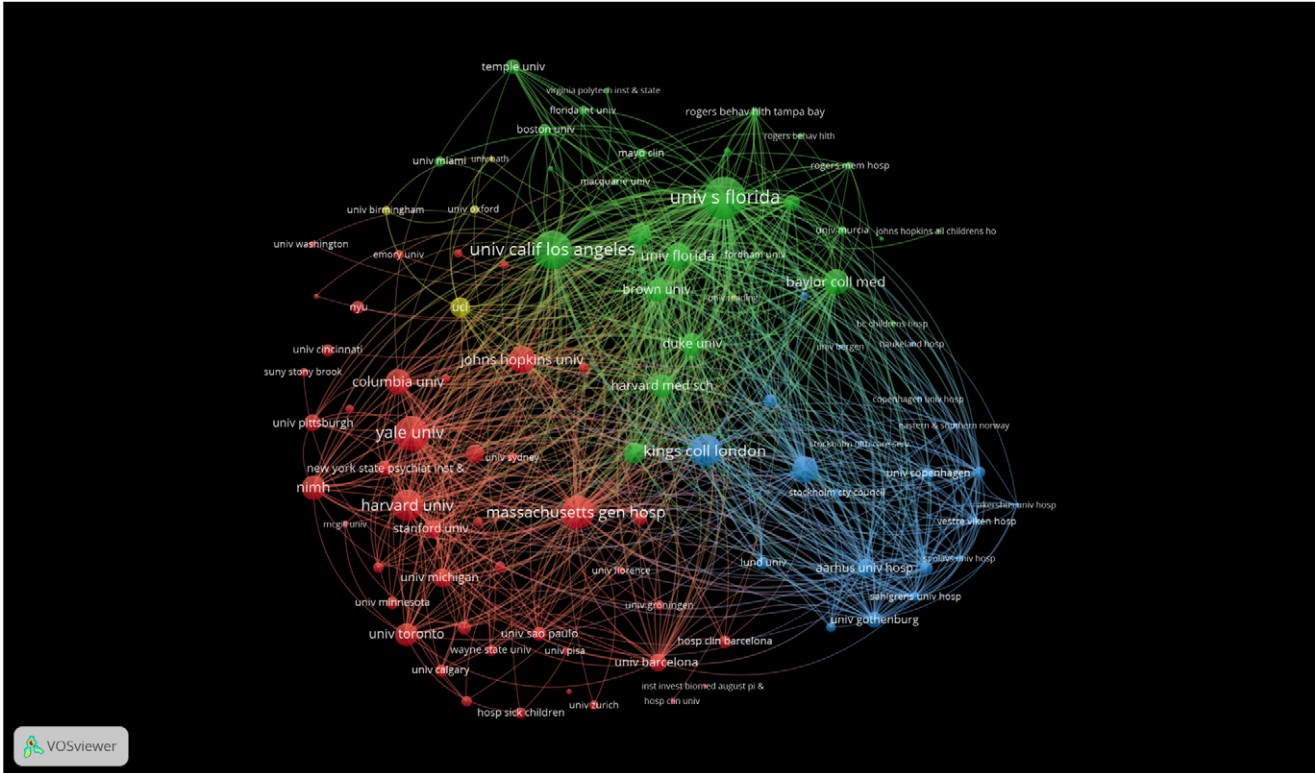

**Figure 5.** Number of publications by institution/organization.

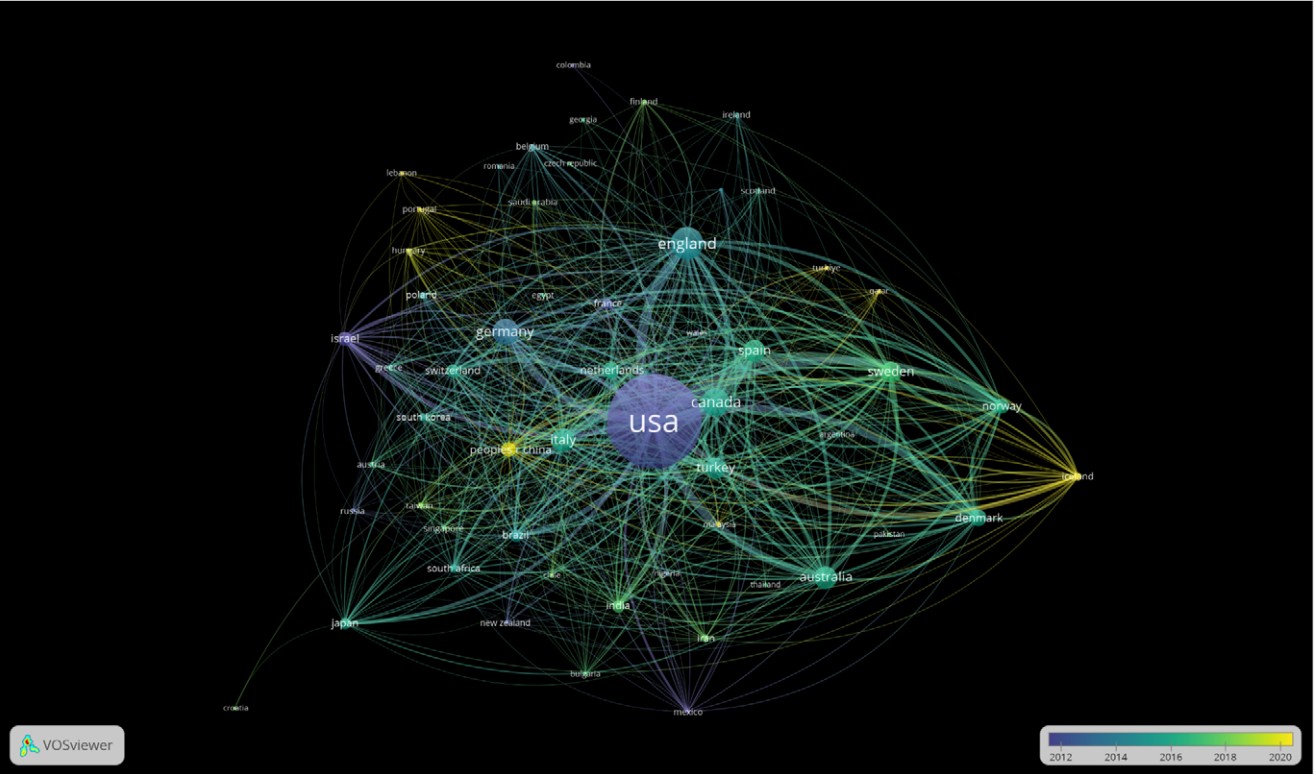

**Figure 6.** Temporal trend of publications.

Canada exhibit close associations with each other in OCD research. In contrast, countries like Portugal and the Nordics tend to have less established connections and produce newer publications. Like with developing countries, this helps provide invaluable insight into OCD research.

## Conclusions

This bibliometric analysis highlights research contributions from both developed and developing countries, showcasing leading contributors like the United States, England and Canada, alongside emerging contributors such as China, India and Turkey. This analysis compares citation and publication strengths, as well as the evolving role of developing nations in OCD research. Incorporating data regarding the cultural nuances influencing OCD research could further expand this review and support the global applicability of the findings. Bibliometric analysis has allowed a view on the development of research in the field of adolescent OCD. As research from developing countries increases, their contributions add valuable data to the global understanding of OCD. The developed world continues to be the most prominent in OCD research; however, developing countries contribute invaluable insight by expanding our understanding, enhancing diversity and increasing representation in the field.

**Open peer review.** To view the open peer review materials for this article, please visit http://doi.org/10.1017/gmh.2025.12.

**Data availability statement.** All data reported in this paper are publicly available from the Web of Science.

**Author contribution.** All authors have made substantial contributions to the conception or design of this work and drafting or revising it critically for important intellectual content and have approved the final version to be published.

**Financial support.** This research received no specific grant from any funding agency, commercial or not-for-profit sectors.

**Competing interest.** The authors declare no competing interests exist.

**Ethics statement.** The data reported in this paper are publicly available, de-identified aggregate data and meet the National Institutes of Health criteria for exempt minimal risk studies.

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
