## [Reviewer Report]

Thank you for giving me an opportunity to review this interesting piece of research.

Overall, the review is very well done and has categorised research on adolescent OCD done in developed nations, developing nations, and organisations. The citation strength, citation and connections have been analysed over a large number of studies. However, there are few points to consider as response to the questions asked. Please see below for the same:

For global reviews, how well does the review cover global content in the inclusion of

research, presentation of results, and/or in the discussion and implications? And how could this be improved/expanded?

The authors have concluded that as the research from developing countries increases, the stigma surrounding OCD in these countries and around the world will decrease. However, there is only instance given of prevalent stigma in Qatar and comparing it to newer research on OCD in recent times. It may be far-reaching to arrive at this conclusion in absence if direct link between increased publication on OCD and stigma reduction in Qatar as well as lack of evidence for other developing countries.

In the abstract, the authors have mentioned that developing countries play an important role in the development of OCD research because of how they deliver different perspectives into

the field given their more distant associations with developed nation’s research. However, the bibliometric analysis doesn’t show any evidence about the different perspectives apart from newer research coming from the developed nations.

For reviews that are regionally focused, how well do the authors describe how the results fit

in with global research and global learnings? And how could this be improved/expanded?

In discussion, authors have noted that United States and Canada are countries that have been considered developed for a very long time, and countries, such as Portugal

and the Nordics developed recently. There is no reference given for this, could the authors validate this statement with some literatures on the timeline.

The authors have differentiated between developed and developing countries. However, it would have also been good to note the differentiations between countries from different geographical locations, like USA and Canada have been noted as closely associated with one another.

---

## [Reviewer Report]

Dear authors,

Thank you very much for the opportunity to review this manuscript.

The manuscript deals with a very interesting topic and has a logical sequence that makes it clear to the reader.

The results are interesting and well presented.

I have a few comments to make the manuscript more robust.

I have 7 comments.

Best regards.

COMMENT 1:

Keywords: The manuscript has great potential to be cited in other studies. I therefore propose adding keywords ‘Bibliometric analysis’, ‘Adolescent’ e ‘Youth populations’. I think that by adding these keywords, the manuscript, if published, could be more detectable in internet search engines and database searches, thus increasing the chances of being cited.

COMMENT 2:

In the ‘Introduction’ section, if possible, put two references instead of just one. If possible. I don’t think it’s obligatory, but it would give the ‘Introduction’ section more substance.

COMMENT 3:

In the ‘Methods’ section it would be good to put a reference to the statement “VosViewer is the principal tool of analysis, with analysis strength being the main method for Association”.

COMMENT 4:

Also in the ‘Methods’ section, it would be good to present the ‘VosViewer’ tool a little better and provide a reference. I suggest that the authors consult this article: https://www.ncbi.nlm.nih.gov/pmc/articles/PMC9782747/

APA Reference: Arruda, H., Silva, E. R., Lessa, M., Proença Jr, D., & Bartholo, R. (2022). VOSviewer and bibliometrix. Journal of the Medical Library Association: JMLA, 110(3), 392.

COMMENT 5:

The pictures in VosViewer need to be improved for better visualisation by the reader.

COMMENT 6:

In the ‘Results’ section where it says ‘1 - Citation strength of each country’, I think I should be ‘Fig. 1 - Citation strength of each country’.

COMMENT 7:

In the ‘Discussion’ section, compare the results with other previous studies, if any.

---

## [Editor Report]

Dear Authors,

Your Manuscript “ Bibliometric analysis of OCD prevalence in youth populations of developing countries” has now been reviewed.

---

## [Reviewer Report]

Dear authors,

I think the authors have responded to the comments adequately.

I have no further comments.

Best regards.

---

## [Editor Report]

Dear Dr Ganti,

Your revised manuscript 'Bibliometric analysis of OCD prevalence in youth populations of developing countries", has now been reviewed.

Discretionary comment regarding the title: Throughout the manuscript, developed countries are included regarding the subject matter( OCD). For example, in the abstract, third sentence states the the most significant research came from developed countries i.e.. US and England. My opinion is that the title does not capture this, it points to developing countries only. I would suggest adding the word ‘Global’ to reflect the content of the manuscript. something like: ‘ Global Bibliometric analysis of OCD prevalence in youth populations: What is the contribution of developing countries?’